# Error Analysis of Tensor-Train Cross Approximation

**Zhen Qin**
Ohio State University
qin.660@osu.edu

**Alexander Lidiak**
Colorado School of Mines
alidiak@mines.edu

**Zhexuan Gong**
Colorado School of Mines
gong@mines.edu

**Gongguo Tang**
University of Colorado
gongguo.tang@colorado.edu

**Michael B. Wakin**
Colorado School of Mines
mwakin@mines.edu

**Zhihui Zhu**[*]
Ohio State University
zhu.3440@osu.edu

## Abstract

Tensor train decomposition is widely used in machine learning and quantum physics due to its concise representation of high-dimensional tensors, overcoming the curse of dimensionality. Cross approximation—originally developed for representing a matrix from a set of selected rows and columns—is an efficient method for constructing a tensor train decomposition of a tensor from few of its entries. While tensor train cross approximation has achieved remarkable performance in practical applications, its theoretical analysis, in particular regarding the error of the approximation, is so far lacking. To our knowledge, existing results only provide element-wise approximation accuracy guarantees, which lead to a very loose bound when extended to the entire tensor. In this paper, we bridge this gap by providing accuracy guarantees in terms of the entire tensor for both exact and noisy measurements. Our results illustrate how the choice of selected subtensors affects the quality of the cross approximation and that the approximation error caused by model error and/or measurement error may not grow exponentially with the order of the tensor. These results are verified by numerical experiments and may have important implications for the usefulness of cross approximations for high-order tensors, such as those encountered in the description of quantum many-body states.

## 1 Introduction

As modern big datasets are typically represented in the form of huge tensors, tensor decomposition has become ubiquitous across science and engineering applications, including signal processing and machine learning [1, 2], communication [3], chemometrics [4, 5], genetic engineering [6], and so on.

The most widely used tensor decompositions include the canonical polyadic (CP) [7], Tucker [8] and tensor train (TT) [9] decompositions. In general, the CP decomposition provides a compact representation but comes with computational difficulties in finding the optimal decomposition for high-order tensors [10, 11]. The Tucker decomposition can be approximately computed via the higher order singular value decomposition (SVD) [12], but it is inapplicable for high-order tensors since its number of parameters scales exponentially with the tensor order. The TT decomposition sits in the middle between CP and Tucker decompositions and combines the best of the two worlds: the number of parameters does not grow exponentially with the tensor order, and it can be approximately computed by the SVD-based algorithms with a guaranteed accuracy [9]. (See the monograph [13] for a detailed description.) As such, TT decomposition has attracted tremendous interest over the past decade [14–26]. Notably, the TT decomposition is equivalent to the *matrix product state (MPS) or matrix product operator (MPO)* introduced in the quantum physics community to efficiently represent

---

[*]Corresponding author.

36th Conference on Neural Information Processing Systems (NeurIPS 2022).

one-dimensional quantum states, where the number of particles in a quantum system determines the order of the tensor [27–29]. The use of TT decomposition may thus allow one to classically simulate a one-dimension quantum many-body system with resources *polynomial* in the number of particles [29], or to perform scalable quantum state tomography [30].

While an SVD-based algorithm [9] can be used to compute a TT decomposition, this requires information about the entire tensor. However, in many practical applications such as low-rank embeddings of visual data [31], density estimation [32], surrogate visualization modeling [33], quantum state tomography [30], multidimensional harmonic retrieval [34], parametric partial differential equations [35], interpolation [36], and CMOS ring oscillators [37], one can or it is desirable to only evaluate a small number of elements of a tensor in practice, since a high-order tensor has exponentially many elements. To address this issue, the work [38] develops a *cross approximation* technique for efficiently computing a TT decomposition from selected subtensors. Originally developed for matrices, cross approximation represents a matrix from a selected set of its rows and columns, preserving structure such as sparsity and non-negativity within the data. Thus, it has been widely used for data analysis [39, 40], subspace clustering [41], active learning [42], approximate SVDs [43, 44], data compression and sampling [45–47], and it has been extended for tensor decompositions [38, 48–52]. See Section 2 for a detailed description of cross approximation for matrices and Section 3 for its extension to TT decomposition.

Although the theoretical analysis and applications of matrix cross approximation [53–64] have been developed for last two decades, to the best of our knowledge, the theoretical analysis of TT cross approximation, especially regarding its error bounds, has not been well studied [65, 66]. Although cross approximation gives an accurate TT decomposition when the tensor is exactly in the TT format [38], the question remains largely unsolved as to how good the approximation is when the tensor is only approximately in the TT format, which is often the case for practical data. In particular, an important question is whether the approximation error grows exponentially with the order due to the cascaded multiplications between the factors; this exponential error growth is proved to occur with the Tucker decomposition [48, 49]. Fortunately, for TT cross approximation, the works [65, 66] establish element-wise error bounds that do not grow exponentially with the order. Unfortunately, when one extends these element-wise results to the entire tensor, the Frobenius norm error bound grows exponentially with the order of the tensor, indicating that the cross approximation could essentially fail for high-order tensors. However, in practice cross approximation is often found to work well even for high-order tensors, leading us to seek for tighter error bounds on the entire tensor. Specifically, we ask the following main question:

> **Question**: Does cross approximation provide a stable tensor train (TT) decomposition of a high-order tensor with accuracy guaranteed in terms of the Frobenius norm?

In this paper, we answer this question affirmatively by developing accuracy guarantees for tensors that are not exactly in the TT format. Our results illustrate how the choice of subtensors affects the quality of the approximation and that the approximation error need not grow exponentially with the order. We also extend the approximation guarantee to noisy measurements.

The rest of this paper is organized as follows. In Section 2, we review and summarize existing error bounds for matrix cross approximation. Section 3 introduces our new error bounds for TT cross approximation. Section 4 includes numerical simulation results that supports our new error bounds and Section 5 concludes the paper.

**Notation**: We use calligraphic letters (e.g., $\mathcal{A}$) to denote tensors, bold capital letters (e.g., $\boldsymbol{A}$) to denote matrices, bold lowercase letters (e.g., $\boldsymbol{a}$) to denote vectors, and italic letters (e.g., $a$) to denote scalar quantities. $\boldsymbol{A}^{-1}$ and $\boldsymbol{A}^{\dagger}$ represent the inverse and the Moore–Penrose pseudoinverse matrices of $\boldsymbol{A}$, respectively. Elements of matrices and tensors are denoted in parentheses, as in Matlab notation. For example, $\mathcal{A}(i_1, i_2, i_3)$ denotes the element in position $(i_1, i_2, i_3)$ of the order-3 tensor $\mathcal{A}$, $\boldsymbol{A}(i_1, :)$ denotes the $i_1$-th row of $\boldsymbol{A}$, and $\boldsymbol{A}(:, J)$ denotes the submatrix of $\boldsymbol{A}$ obtained by taking the columns indexed by $J$. $\|\boldsymbol{A}\|$ and $\|\boldsymbol{A}\|_F$ respectively represent the spectral norm and Frobenius norm of $\boldsymbol{A}$.

## 2 Cross Approximation for Matrices

As a tensor can be viewed as a generalization of a matrix, it is instructive to introduce the principles behind cross approximation for matrices. Cross approximation, also known as *skeleton decomposition* or *CUR decomposition*, approximates a low-rank matrix using a small number of its rows and columns. Formally, following the conventional notation for the CUR decomposition, we construct a cross approximation for any matrix $A \in \mathbb{R}^{m \times n}$ as

$$A \approx CU^{\dagger}R, \tag{1}$$

where $\dagger$ denotes the pseudoinverse (not to be confused with the Hermitian conjugate widely used in quantum physics literature), $C = A(:, J)$, $U = A(I, J)$, and $R = A(I, :)$ with $I \in [m]$ and $J \in [n]$ denoting the indices of the selected rows and columns, respectively. See Figure 1 for an illustration of matrix cross approximation. While the number of selected rows and columns (i.e., $|I|$ and $|J|$) can be different, they are often the same since the rank of $CU^{\dagger}R$ is limited by the smaller number if they are different. When $U$ is square and non-singular, the cross approximation (1) takes the standard form $A \approx CU^{-1}R$. Without loss of generality, we assume an equal number of selected rows and columns in the sequel. We note that one may replace $U^{\dagger}$ in (1) by $C^{\dagger}AR^{\dagger}$ to improve the approximation, resulting in the CUR approximation $CC^{\dagger}AR^{\dagger}R$, where $CC^{\dagger}$ and $R^{\dagger}R$ are orthogonal projections onto the subspaces spanned by the given columns and rows, respectively [67, 61]. However, such a CUR approximation depends on knowledge of the entire matrix $A$. As we mainly focus on scenarios where such knowledge is prohibitive, in the sequel we only consider the cross approximation (1) with $U = A(I, J)$.

While in general the singular value decomposition (SVD) gives the best low-rank approximation to a matrix in terms of Frobenius norm or spectral norm, the cross approximation shown in Figure 1 has several notable advantages: $(i)$ it is more interpretable since it is constructed

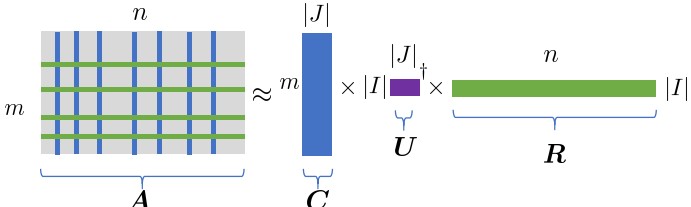

Figure 1: Cross approximation for a matrix.

with the rows and columns of the matrix, and hence could preserve structure such as sparsity, non-negativity, etc., and $(ii)$ it only requires selected rows and columns while other factorizations often require the entire matrix. As such, cross approximation has been widely used for data analysis [39, 40], subspace clustering [41], active learning [42], approximate SVDs [43, 44], data compression and sampling [45–47], and so on; see [61, 62] for a detailed history of the use of cross approximation.

The following result shows that the cross approximation is exact (i.e., (1) becomes equality) as long as the submatrix $U$ has the same rank as $A$.

**Theorem 1** *For any $A \in \mathbb{R}^{m \times n}$, suppose we select rows and columns $C = A(:, J)$, $U = A(I, J)$, $R = A(I, :)$ from $A$ such that* $\mathrm{rank}(U) = \mathrm{rank}(A)$. *Then*

$$A = CU^{\dagger}R.$$

**Proof** As $R$ is a submatrix of $A$ and $U$ is a submatrix of $R$, $\mathrm{rank}(U) \leq \mathrm{rank}(R) \leq \mathrm{rank}(A)$ always holds. Now if $\mathrm{rank}(U) = \mathrm{rank}(A)$, we have $\mathrm{rank}(U) = \mathrm{rank}(R)$ and thus any columns in $R$ can be linearly represented by the columns in $U$. Using this observation and noting that $U = UU^{\dagger}U$, we can obtain that $R = UU^{\dagger}R$. Likewise, since $\mathrm{rank}(R) = \mathrm{rank}(A)$, any rows in $A$ can be linearly represented by the rows in $R$. Thus, we can conclude that $A = CU^{\dagger}R$. ∎

In words, Theorem 1 implies that for an exactly low-rank matrix $A$ with rank $r$, we can sample $r$ (or more) rows and columns to ensure exact cross approximation. In practice, however, $A$ is generally not exactly low-rank but only approximately low-rank. In this case, the performance of cross approximation depends crucially on the selected rows and columns. Column subset selection is a classical problem in numerical linear algebra, and many methods have been proposed based on the maximal volume principle [68, 69] or related ideas [70–72]; see [72] for more references representing this research direction.

Efforts have also been devoted to understand the robustness of cross approximation since it only yields a low-rank approximation to $\boldsymbol{A}$ when $\boldsymbol{A}$ is not low-rank. The work [53] provides an element-wise accuracy guarantee for the rows and columns selected via the maximal volume principle. The recent work [58] shows there exist a stable cross approximation with accuracy guaranteed in the Frobenius norm, while the work [59] establishes a similar guarantee for cross approximations of random matrices using the maximum projective volume principle. A derandomized algorithm is proposed in [60] that finds a cross approximation with a similar performance guarantee. The work [62] establishes an approximation guarantee for cross approximation using any sets of selected rows and columns. See Appendix A for a detailed description of these results.

# 3 Cross Approximation for Tensor Train Decomposition

In this section, we consider cross approximation for tensor train (TT) decomposition. We begin by introducing tensor notations. For an $N$-th order tensor $\mathcal{T} \in \mathbb{R}^{d_1 \times \cdots \times d_N}$, the $k$-th *separation* or *unfolding* of $\mathcal{T}$, denoted by $\boldsymbol{T}^{\langle k \rangle}$, is a matrix of size $(d_1 \cdots d_k) \times (d_{k+1} \cdots d_N)$ with its $(i_1 \ldots i_k, i_{k+1} \cdots i_N)$-th element defined by

$$\boldsymbol{T}^{\langle k \rangle}(i_1 \cdots i_k, i_{k+1} \cdots i_N) = \mathcal{T}(i_1, \ldots, i_N). \tag{2}$$

Here, $i_1 \cdots i_k$ is a single multi-index that combines indices $i_1, \ldots, i_k$. As for the matrix case, we will select rows and columns from $\boldsymbol{T}^{\langle k \rangle}$ to construct a cross approximation for all $k = 1, \ldots, N-1$. To simplify the notation, we use $I^{\leq k}$ and $I^{>k}$ to denote the positions of the selected $r'_k$ rows and columns in the $k$-th unfolding. With abuse of notation, $I^{\leq k}$ and $I^{>k}$ are also used as the positions in the original tensor. Thus, both $\boldsymbol{T}^{\langle k \rangle}(I^{\leq k}, I^{>k})$ and $\boldsymbol{T}(I^{\leq k}, I^{>k})$ represent the sampled $r'_k \times r'_k$ intersection matrix. Likewise, $\boldsymbol{T}(I^{\leq k-1}, i_k, I^{>k})$ represents an $r'_{k-1} \times r'_k$ matrix whose $(s, t)$-th element is given by $\boldsymbol{T}(I_s^{\leq k-1}, i_k, I_t^{>k})$ for $s = 1, \ldots, r'_{k-1}$ and $t = 1, \ldots, r'_k$.

## 3.1 Tensor Train Decomposition

We say that $\mathcal{T} \in \mathbb{R}^{d_1 \times \cdots \times d_N}$ is in the *TT format* if we can express its $(i_1, \ldots, i_N)$-th element as the following matrix product form [9]

$$\mathcal{T}(i_1, \ldots, i_N) = \boldsymbol{X}_1(:, i_1, :) \cdots \boldsymbol{X}_N(:, i_N, :), \tag{3}$$

where $\boldsymbol{X}_j \in \mathbb{R}^{r_{j-1} \times d_j \times r_j}, j = 1, \ldots, N$ with $r_0 = r_N = 1$ and $\boldsymbol{X}_j(:, i_j, :) \in \mathbb{R}^{r_{j-1} \times r_j}, j = 1, \ldots, N$ denotes one "slice" of $\boldsymbol{X}_j$ with the second index being fixed at $i_j$. To simplify notation, we often write the above TT format in short as $\mathcal{T} = [\boldsymbol{X}_1, \ldots, \boldsymbol{X}_N]$. In quantum many-body physics, (3) is equivalent to a matrix product state or matrix product operator, where $\mathcal{T}(i_1, \ldots, i_N)$ denotes an element of the many-body wavefunction or density matrix, $N$ is the number of spins/qudits, and $r_k$ ($k = 1, 2, \cdots, N-1$) are known as the bond dimensions. Note that in general the decomposition (3) is not invariant to dimensional permutation due to strictly sequential multilinear products over the latent cores. The tensor ring decomposition generalizes (3) by taking the trace of the RHS; this allows $r_0, r_N \geq 1$ and is invariant to circular dimensional permutation [73].

While there may exist infinitely many ways to decompose a tensor $\mathcal{T}$ as in (3), we say the decomposition is *minimal* if the rank of the left unfolding of each $\boldsymbol{X}_k$ (i.e., $L(\boldsymbol{X}_k) = \begin{bmatrix} \boldsymbol{X}_k(:, 1, :) \\ \vdots \\ \boldsymbol{X}_k(:, d_k, :) \end{bmatrix}$) is $r_k$ and the rank of the right unfolding of each $\boldsymbol{X}_k$ (i.e., $R(\boldsymbol{X}_k) = [\boldsymbol{X}_k(:, 1, :) \quad \cdots \quad \boldsymbol{X}_k(:, d_k, :)]$) is $r_{k-1}$. The dimensions $\boldsymbol{r} = (r_1, \ldots, r_{N-1})$ of such a minimal decomposition are called the *TT ranks* of $\mathcal{T}$. According to [74], there is exactly one set of ranks $\boldsymbol{r}$ that $\mathcal{T}$ admits a minimal TT decomposition. Moreover, $r_k$ equals the rank of the $k$-th unfolding matrix $\boldsymbol{T}^{\langle k \rangle}$, which can serve as an alternative way to define the TT rank.

**Efficiency of Tensor Train Decomposition**   The number of elements in the $N$-th order tensor $\mathcal{T}$ grows exponentially in $N$, a phenomenon known as the *curse of dimensionality*. In contrast, a TT decomposition of the form (3) can represent a tensor $\mathcal{T}$ with $d_1 \cdots d_N$ elements using only $O(dNr^2)$ elements, where $d = \max\{d_1, \ldots, d_N\}$ and $r = \max\{r_1, \ldots, r_{N-1}\}$. This makes the TT decomposition remarkably efficient in addressing the curse of dimensionality as its number of

parameters scales only linearly in $N$. For this reason, the TT decomposition has been widely used for image compression [14, 15], analyzing theoretical properties of deep networks [16], network compression (or tensor networks) [17–22], recommendation systems [23], probabilistic model estimation [24], learning of Hidden Markov Models [25], and so on. The work [26] presents a library for TT decomposition based on TensorFlow. Notably, TT decomposition is equivalent to the *matrix product state (MPS)* introduced in the quantum physics community to efficiently and concisely represent quantum states; here $N$ represents the number of qubits of the many–body system [27–29]. The concise representation by MPS is remarkably useful in quantum state tomography since it allows us to observe a quantum state with both experimental and computational resources that are only *polynomial* rather than *exponential* in the number of qubits $N$ [30].

**Comparison with Other Tensor Decompositions**   Other commonly used tensor decompositions include the Tucker decomposition, CP decomposition, etc. The Tucker decomposition is suitable only for small-order tensors (such as $N = 3$) since its parameterization scales exponentially with $N$ [8]. Like the TT decomposition, the canonical polyadic (CP) decomposition [7] also represents a tensor with a linear (in $N$) number of parameters. However, TT has been shown to be exponentially more expressive than CP for almost any tensor [16, 75]. Moreover, computing the CP decomposition is computationally challenging; even computing the canonical rank can be NP-hard and ill-posed [10, 11]. In contrast, one can use a sequential SVD-based algorithm [9] to compute a quasioptimal TT decomposition (the accuracy differs from that of the best possible approximation by at most a factor of $\sqrt{N}$). We refer to [13] for a detailed comparison.

### 3.2 Cross Approximation For Tensor Train Decomposition

While the SVD-based algorithm [9] can be used to find a TT decomposition, that algorithm requires the entire tensor and is inefficient for large tensor order $N$. More importantly, in applications like low-rank embeddings of visual data [31], density estimation [32], surrogate visualization modeling [33], and quantum state tomography [30], it is desirable to measure a small number of tensor elements since the experimental cost is proportional to the number of elements measured. In such scenarios, we can use TT cross approximation to reconstruct the full tensor using only a small number of known elements, similar to the spirit of matrix cross approximation.

To illustrate the main ideas behind TT cross approximation, we first consider the case where a tensor $\mathcal{T}$ is in the exact TT format (3), i.e., the unfolding matrix $\boldsymbol{T}^{\langle k \rangle}$ is exactly low-rank with rank $r_k$. In this case, by Theorem 1, we can represent $\boldsymbol{T}^{\langle 1 \rangle}$ with a cross approximation by choosing $r_1'$ rows (with $r_1' \geq r_1$) indexed by $I^{\leq 1}$ and $r_1'$ columns indexed by $I^{>1}$ such that

$$\boldsymbol{T}^{\langle 1 \rangle} = \boldsymbol{T}^{\langle 1 \rangle}(:, I^{>1})[\boldsymbol{T}^{\langle 1 \rangle}(I^{\leq 1}, I^{>1})]^\dagger \boldsymbol{T}^{\langle 1 \rangle}(I^{\leq 1}, :).$$

Here $\boldsymbol{T}^{\langle 1 \rangle}(:, I^{>1})$ has $d_1 r_1'$ entries, while $\boldsymbol{T}^{\langle 1 \rangle}(I^{\leq 1}, :)$ is an extremely wide matrix. However, *one only needs to sample the elements in* $\boldsymbol{T}^{\langle 1 \rangle}(:, I^{>1})$ *and can avoid sampling the entire elements of* $\boldsymbol{T}^{\langle 1 \rangle}(I^{\leq 1}, :)$ *by further using cross approximation.* In particular, one can reshape $\boldsymbol{T}^{\langle 1 \rangle}(I^{\leq 1}, :)$ into another matrix of size $r_1' d_2 \times d_3 \cdots d_N$ and then represent it with a cross approximation by choosing $r_2'$ rows and columns. This procedure can be applied recursively $N - 1$ times, and only at the last

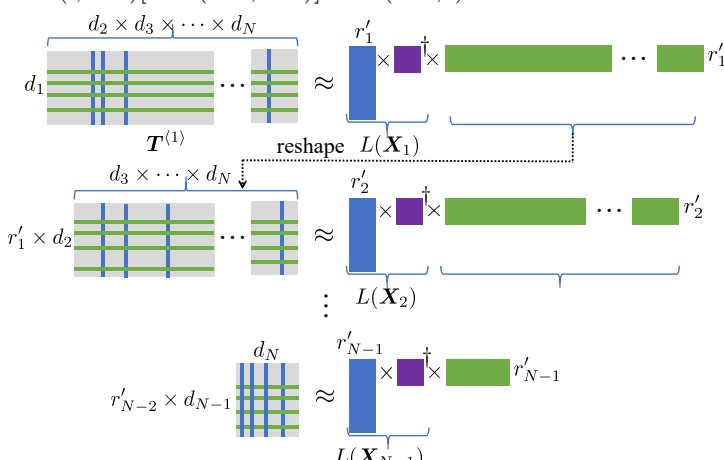

Figure 2: Cross approximation for TT format tensors with nested interpolation sets.

stage does one sample both $r_{N-1}'$ rows and columns of an $r_{N-2}' d_{N-1} \times d_N$ matrix. This is the procedure developed in [38] for TT cross approximation. The process of TT cross approximation is shown in Fig. 2.

The procedure illustrated in Figure 2 produces nested interpolation sets as one recursively applies cross approximation on the reshaped versions of previously selected rows. However, the interpolation sets $\{I^{\leq k}, I^{>k}\}$ are not required to be nested. In general, one can slightly generalize this procedure and obtain a TT cross approximation $\widehat{\mathcal{T}}$ with elements given by [66]

$$\widehat{\mathcal{T}}(i_1,\ldots,i_N) = \prod_{k=1}^{N} \boldsymbol{T}(I^{\leq k-1}, i_k, I^{>k})[\boldsymbol{T}(I^{\leq k}, I^{>k})]^{\dagger}_{\tau_k}, \tag{4}$$

where $I^{\leq k}$ denotes the selected row indices from the multi-index $\{d_1 \cdots d_k\}$, $I^{>k}$ is the selection of the column indices from the multi-index $\{d_{k+1} \cdots d_N\}$, and $I^{\leq 0} = I^{>N} = \emptyset$. Here $\boldsymbol{A}^{\dagger}_{\tau}$ denotes the $\tau$-pseudoinversion of $\boldsymbol{A}$ that sets the singular values less than $\tau$ to zero before calculating the pseudoinversion.

Similar to Theorem 1, the TT cross approximation $\widehat{\mathcal{T}}$ equals $\mathcal{T}$ when $\mathcal{T}$ is in the TT format (3) and the intersection matrix $\boldsymbol{T}(I^{\leq k}, I^{>k})$ has rank $r_k$ for $k = 1, \ldots, N-1$ [38]. Nevertheless, in practical applications, tensors may be only approximately but not exactly in the TT format (3), i.e., the unfolding matrices $\boldsymbol{T}^{\langle k \rangle}$ may be only approximately low-rank. Moreover, in applications such as quantum tomography, one can only observe noisy samples of the tensor entries. We provide guarantees in terms of $\|\mathcal{T} - \widehat{\mathcal{T}}\|_F$ for these cases in the following subsections.

### 3.3 Error Analysis of TT Cross Approximation with Exact Measurements

Assume each unfolding matrix $\boldsymbol{T}^{\langle k \rangle}$ is approximately low-rank with residual captured by $\|\boldsymbol{T}^{\langle k \rangle} - \boldsymbol{T}^{\langle k \rangle}_{r_k}\|_F$, where $\boldsymbol{T}^{\langle k \rangle}_{r_k}$ denotes the best rank-$r_k$ approximation of $\boldsymbol{T}^{\langle k \rangle}$. For convenience, we define

$$r = \max_{k=1,\ldots,N-1} r_k, \quad \epsilon = \max_{k=1,\ldots,N-1} \|\boldsymbol{T}^{\langle k \rangle} - \boldsymbol{T}^{\langle k \rangle}_{r_k}\|_F, \tag{5}$$

where $\epsilon$ captures the largest low-rank approximation error in all the unfolding matrices.

The work [65, 66] develops an element-wise error bound for TT cross approximation based on the maximum-volume choice of the interpolation sets. However, if we are interested in an approximation guarantee for the entire tensor (say in the Frobenius norm) instead of each entry, then directly applying the above result leads to a loose bound with scaling at least proportional to $\sqrt{d_1 \cdots d_N}$. Our goal is to provide guarantees directly for $\|\mathcal{T} - \widehat{\mathcal{T}}\|_F$. Note that (5) implies that a fundamental lower bound

$$\|\mathcal{T} - \widehat{\mathcal{T}}\|_F \geq \epsilon \quad \text{since} \quad \|\mathcal{T} - \widehat{\mathcal{T}}\|_F = \|\boldsymbol{T}^{\langle k \rangle} - \widehat{\boldsymbol{T}}^{\langle k \rangle}\|_F \geq \|\boldsymbol{T}^{\langle k \rangle} - \boldsymbol{T}^{\langle k \rangle}_{r_k}\|_F \tag{6}$$

holds for *any* TT format tensor $\widehat{\mathcal{T}}$ with ranks $\boldsymbol{r}$ (not only those constructed by cross approximation). Fortunately, tensor models in practical applications such as image and video [14, 15] and quantum states [27–29] can be well represented in the TT format with very small $\epsilon$.

Our first main result proves the existence of stable TT cross approximation with exact measurements of the tensor elements:

**Theorem 2** *Suppose $\mathcal{T}$ can be approximated by a TT format tensor with ranks $\boldsymbol{r}$ and approximation error $\epsilon$ defined in (5). For sufficiently small $\epsilon$, there exists a cross approximation $\widehat{\mathcal{T}}$ of the form (4) with $\tau_k = 0$ that satisfies*

$$\left\|\mathcal{T} - \widehat{\mathcal{T}}\right\|_F \leq \frac{(3\kappa)^{\lceil \log_2 N \rceil} - 1}{3\kappa - 1}(r+1)\epsilon, \tag{7}$$

*where $\kappa := \max_k \left\{ \left\|\boldsymbol{T}^{\langle k \rangle}(:, I^{>k}) \cdot \boldsymbol{T}^{\langle k \rangle}(I^{\leq k}, I^{>k})^{-1}\right\|, \left\|\boldsymbol{T}^{\langle k \rangle}(I^{\leq k}, I^{>k})^{-1} \cdot \boldsymbol{T}^{\langle k \rangle}(I^{\leq k}, :)\right\| \right\}$.*

We use sufficiently small $\epsilon$ to simplify the presentation. This refers to the value of $\epsilon$ that $\left\|\left(\widehat{\boldsymbol{T}}^{\langle k \rangle}(:, I^{>k}) - \boldsymbol{T}^{\langle k \rangle}(:, I^{>k})\right) \cdot \boldsymbol{T}^{\langle k \rangle}(I^{\leq k}, I^{>k})^{-1}\right\| \leq \kappa$, which can always be guaranteed since by (7) the distance between $\widehat{\mathcal{T}}$ and $\mathcal{T}$ (and hence the distance between $\widehat{\boldsymbol{T}}^{\langle k \rangle}(:, I^{>k})$ and $\boldsymbol{T}^{\langle k \rangle}(:, I^{>k})$) approaches zero when $\epsilon$ goes to zero. The full proof is provided in Appendix B.

On one hand, the right hand side of (7) grows only linearly with the approximation error $\epsilon$, with a scalar that increases only logarithmically in terms of the order $N$. This shows the TT cross approximation

can be stable for well selected interpolation sets. Recall that by (6), for any cross approximation $\widehat{\mathcal{T}}$ of the form (4) with $r_k$ selected rows and columns in the sets $\{I^{\leq k}, I^{>k}\}$, $\|\mathcal{T} - \widehat{\mathcal{T}}\|_F \geq \epsilon$ necessarily holds since $\widehat{\boldsymbol{T}}^{\langle k \rangle}$ has rank at most $r_k$. While it may be unfair to compare the element-wise error bounds [65, 66], we present [65, Theorem 1] below which also involves $\epsilon$ and a similar quantity to $\kappa$:

$$\|\mathcal{T} - \widehat{\mathcal{T}}\|_{\max} \leq (2r + r\widetilde{\kappa} + 1)^{\lceil \log_2 N \rceil}(r+1)\epsilon, \quad \text{where } \|\mathcal{T}\|_{\max} = \max_{i_1, \ldots, i_N} |\mathcal{T}(i_1, \ldots, i_N)|$$

and $\widetilde{\kappa} = \max_k r_k \|\boldsymbol{T}^{\langle k \rangle}\|_{\max} \cdot \|\boldsymbol{T}^{\langle k \rangle}(I^{\leq k}, I^{>k})^{-1}\|_{\max}$. Loosely speaking, $\kappa$ and $\widetilde{\kappa}$ can be approximately viewed as the condition number of the submatrix $\boldsymbol{T}^{\langle k \rangle}(I^{\leq k}, I^{>k})$ with respect to the spectral norm and the Chebyshev norm, respectively. Again, we note that our bound (7) is for the entire tensor, while $\|\mathcal{T} - \widehat{\mathcal{T}}\|_{\max}$ only concerns the largest element-wise error.

On the other hand, Theorem 2 only shows the existence of one such stable cross approximation, but it does not specify which one. As a consequence, the result may not hold for the nested sampling strategy illustrated in Figure 2 and the parameter $\kappa$ maybe out of one's control. The following result addresses these issues by providing guarantees for any cross approximation.

**Theorem 3** *Suppose $\mathcal{T}$ can be approximated by a TT format tensor with rank $r$ and approximation error $\epsilon$ defined in (5). Let $\boldsymbol{T}_{r_k}^{\langle k \rangle}$ be the best rank $r_k$ approximation of the $k$-th unfolding matrix $\boldsymbol{T}^{\langle k \rangle}$ and $\boldsymbol{T}_{r_k}^{\langle k \rangle} = \boldsymbol{W}_{(k)} \boldsymbol{\Sigma}_{(k)} \boldsymbol{V}_{(k)}^T$ be its compact SVD. For any interpolation sets $\{I^{\leq k}, I^{>k}\}$ such that $\mathrm{rank}(\boldsymbol{T}_{r_k}^{\langle k \rangle}(I^{\leq k}, I^{>k})) = r_k, k = 1, \ldots, N-1$, denote by*

$$a = \max_{k=1,\ldots,N-1} \left\{ \left\| [\boldsymbol{W}_{(k)}(I^{\leq k}, :)]^\dagger \right\|, \left\| [\boldsymbol{V}_{(k)}(I^{>k}, :)]^\dagger \right\| \right\}, \quad c = \max_{k=1,\ldots,N-1} \left\| [\boldsymbol{T}^{\langle k \rangle}(I^{\leq k}, I^{>k})]^\dagger \right\|.$$

*Then the cross approximation $\widehat{\mathcal{T}}$ in (4) with appropriate thresholding parameters $\tau_k$ for the truncated pseudo-inverse satisfies*

$$\|\mathcal{T} - \widehat{\mathcal{T}}\|_F \lesssim (a^2 r + a^2 cr\epsilon + a^2 c^2 \epsilon^2)^{\lceil \log_2 N \rceil - 1} \cdot (a^2 \epsilon + a^2 c\epsilon^2 + a^2 c^2 \epsilon^3), \quad (8)$$

*where $\lesssim$ means less than or equal to up to some universal constant.*

The proof of Theorem 3 is given in Appendix C. As in (7), the right hand side of (8) scales only polynomially in $\epsilon$ and logarithmically in terms of the order $N$. Compared with (7), the guarantee in (8) holds for any cross approximation, but the quality of the cross approximation depends on the tensor $\mathcal{T}$ as well as the selected rows and columns as reflected by the parameters $a$ and $c$. On one hand, $\left\| [\boldsymbol{W}_{(k)}(I^{\leq k}, :)]^\dagger \right\|$ and $\left\| [\boldsymbol{V}_{(k)}(I^{>k}, :)]^\dagger \right\|$ can achieve their smallest possible value 1 when the singular vectors $\boldsymbol{W}_{(k)}$ and $\boldsymbol{V}_{(k)}$ are the canonical basis. On the other hand, one may construct examples with large $\left\| [\boldsymbol{W}_{(k)}(I^{\leq k}, :)]^\dagger \right\|$ and $\left\| [\boldsymbol{V}_{(k)}(I^{>k}, :)]^\dagger \right\|$. Nevertheless, these terms can be upper bounded by selecting appropriate rows and columns. For example, for any orthonormal matrix $\boldsymbol{W} \in \mathbb{R}^{m \times r}$, if we select $I$ such that $\boldsymbol{W}(I, :)$ has maximal volume among all $|I| \times r$ submatrices of $\boldsymbol{W}$, then $\left\| \boldsymbol{W}(I, :)^\dagger \right\| \leq \sqrt{1 + \frac{r(m-|I|)}{|I|-r+1}}$ [62, Proposition 5.1]. In practice, without knowing $\boldsymbol{W}_{(k)}$ and $\boldsymbol{V}_{(k)}$, we may instead find the interpolation sets by maximizing the volume of $\boldsymbol{T}^{\langle k \rangle}(I^{\leq k}, I^{>k})$ using algorithms such as greedy restricted cross interpolation (GRCI) [65]. We may also exploit derandomized algorithms [60] to find interpolation sets that could potentially yield small quantities $a, b, c$. We finally note that Theorem 3 uses the truncated pseudo-inverse in (4) with appropriate $\tau_k$. We also observe from the experiments in Section 4 that the truncated pseudo-inverse could lead to better performance than the inverse (or pseudo-inverse) for TT cross approximation (4).

### 3.4 Error Analysis for TT Cross Approximation with Noisy Measurements

We now consider the case where the measurements of tensor elements are not exact, but noisy. This happens, for example, in the measurements of a quantum state where each entry of the many-body wavefunction or density matrix can only be measured up to some statistical error that depends on the number of repeated measurements. In this case, the measured entries $\boldsymbol{T}(I^{\leq k-1}, i_k, I^{>k})$ and $[\boldsymbol{T}(I^{\leq k}, I^{>k})]_{\tau_k}$ in the cross approximation (4) are noisy. Our goal is to understand how such noisy measurements will affect the quality of cross approximation.

To simplify the notation, we let $\mathcal{E}$ denote the measurement error though it has non-zero values only at the selected interpolation sets $\{I^{\leq k}, I^{>k}\}$. Also let $\widetilde{\mathcal{T}} = \mathcal{T} + \mathcal{E}$ and $\widetilde{\boldsymbol{T}}^{\langle k \rangle} = \boldsymbol{T}^{\langle k \rangle} + \boldsymbol{E}^{\langle k \rangle}$, where $\boldsymbol{E}^{\langle k \rangle}$

denotes the $k$-th unfolding matrix of the noise. Then, the cross approximation of $\mathcal{T} \in \mathbb{R}^{d_1 \times \cdots \times d_N}$ with noisy measurements, denoted by $\widehat{\mathcal{T}}$, has entries given by

$$\widehat{\mathcal{T}}(i_1, \ldots, i_N) = \prod_{k=1}^{N} \widetilde{\boldsymbol{T}}(I^{\leq k-1}, i_k, I^{>k})[\widetilde{\boldsymbol{T}}(I^{\leq k}, I^{>k})]_{\tau_k}^{\dagger}. \tag{9}$$

One can view $\widehat{\mathcal{T}}$ as a cross approximation of $\widetilde{\mathcal{T}}$ with exact measurements of $\widetilde{\mathcal{T}}$. However, we cannot directly apply Theorem 3 since it would give a bound for $\|\widetilde{\mathcal{T}} - \widehat{\mathcal{T}}\|_F$ and that would also depend on the singular vectors of $\widetilde{\boldsymbol{T}}^{\langle k \rangle}$. The following result addresses this issue.

**Theorem 4** *Suppose $\mathcal{T}$ can be approximated by a TT format tensor with rank $r$ and approximation error $\epsilon$ defined in (5). Let $\boldsymbol{T}_{r_k}^{\langle k \rangle}$ be the best rank $r_k$ approximation of the $k$-th unfolding matrix $\boldsymbol{T}^{\langle k \rangle}$ and $\boldsymbol{T}_{r_k}^{\langle k \rangle} = \boldsymbol{W}_{(k)} \boldsymbol{\Sigma}_{(k)} \boldsymbol{V}_{(k)}^T$ be its compact SVD. Let $\widetilde{\mathcal{T}} = \mathcal{T} + \mathcal{E}$ denote the noisy tensor where $\mathcal{E}$ represents the measurement error. For any interpolation sets $\{I^{\leq k}, I^{>k}\}$ such that $\mathrm{rank}(\boldsymbol{T}_{r_k}^{\langle k \rangle}(I^{\leq k}, I^{>k})) = r_k, k = 1, \ldots, N-1$, denote by*

$$a = \max_{k=1,\ldots,N-1} \left\{ \left\| [\boldsymbol{W}_{(k)}(I^{\leq k}, :)]^{\dagger} \right\|, \left\| [\boldsymbol{V}_{(k)}(I^{>k}, :)]^{\dagger} \right\| \right\}, \quad c = \max_{k=1,\ldots,N-1} \left\| [\widetilde{\boldsymbol{T}}^{\langle k \rangle}(I^{\leq k}, I^{>k})]^{\dagger} \right\|.$$

*Denote by $\bar{\epsilon} = \epsilon + \|\mathcal{E}\|_F$. Then the noisy cross approximation $\widehat{\mathcal{T}}$ in (9) with appropriate thresholding parameters $\tau_k$ for the truncated pseudo-inverse satisfies*

$$\|\mathcal{T} - \widehat{\mathcal{T}}\|_F \quad \lesssim \quad (a^2 r + a^2 c r \bar{\epsilon} + a^2 c^2 \bar{\epsilon}^2)^{\lceil \log_2 N \rceil - 1} \cdot \left( a^2 \bar{\epsilon} + a^2 c \bar{\epsilon}^2 + a^2 c^2 \bar{\epsilon}^2 \right). \tag{10}$$

The proof of Theorem 4 is given in Appendix D. In words, Theorem 4 provides a similar guarantee to that in Theorem 3 for TT cross approximation with noisy measurements and shows that it is also stable with respect to measurement error. The parameter $c$ in Theorem 4 depends on the spectral norm of $[\widetilde{\boldsymbol{T}}^{\langle k \rangle}(I^{\leq k}, I^{>k})]^{\dagger}$ and is defined in this form for simplicity. It can be bounded by $[\boldsymbol{T}^{\langle k \rangle}(I^{\leq k}, I^{>k})]^{\dagger}$ and the noise and is expected to be close to $[\boldsymbol{T}^{\langle k \rangle}(I^{\leq k}, I^{>k})]^{\dagger}$ for random noise. We finally note that the measurement error $\mathcal{E}$ only has $O(Nr^2)$ non-zero elements, and thus $\|\mathcal{E}\|_F$ is not exponentially large in terms of $N$.

## 4 Numerical Experiments

In this section, we conduct numerical experiments to evaluate the performance of the cross approximation (9) with noisy measurements. We generate an $N$-th order tensor $\mathcal{T} \in \mathbb{R}^{d_1 \times \cdots \times d_N}$ approximately in the TT format as $\mathcal{T} = \mathcal{T}_r + \eta \mathcal{F}$, where $\mathcal{T}_r$ is in the TT format with mode ranks $[r_1, \ldots, r_{N-1}]$, which is generated from truncating a random Gaussian tensor using a sequential SVD [9], and $\mathcal{F}$ is a random tensor with independent entries generated from the normal distribution. We then normalize $\mathcal{T}_r$ and $\mathcal{F}$ to unit Frobenius norm. Thus $\eta$ controls the low-rank approximation error. To simplify the selection of parameters, we let $d = d_1 = \cdots = d_N = 2, r = r_1 = \cdots = r_{N-1}$, and $\tau = \tau_1 = \cdots = \tau_{N-1}$ for the cross approximation (9).

We apply the greedy restricted cross interpolation (GRCI) algorithm in [65] for choosing the interpolation sets $I^{\leq k}$ and $I^{>k}$, $k = 1, \ldots, N$. For convenience, we set $r' = r'_1 = \cdots = r'_{N-1}$ for the number of selected rows and columns indexed by $I^{\leq k}$ and $I^{>k}$. During the procedure of cross approximation, for each of the $(N-2)dr'^2 + 2dr' - r'^2$ observed entries, we add measurement error randomly generated from the Gaussian distribution of mean 0 and variance $\frac{1}{(N-2)dr'^2 + 2dr' - r'^2}$. As in Section 3.4, for convenience, we model the measurement error as a tensor $\mathcal{E}$ such that $\mathbb{E} \|\mathcal{E}\|_F^2 = 1$. To control the signal-to-noise level, we scale the noise by a factor $\mu$. Thus, we have

$$\widetilde{\mathcal{T}} = \mathcal{T} + \mu \mathcal{E} = \mathcal{T}_r + \eta \mathcal{F} + \mu \mathcal{E}, \tag{11}$$

where $\|\mathcal{T}_r\|_F^2 = \|\mathcal{F}\|_F^2 = \mathbb{E} \|\mathcal{E}\|_F^2 = 1$.

We measure the normalized mean square error (MSE) between $\mathcal{T}$ and the cross approximation $\widehat{\mathcal{T}}$ by

$$\mathrm{MSE} = 10 \log_{10} \frac{\|\mathcal{T} - \widehat{\mathcal{T}}\|_F^2}{\|\mathcal{T}\|_F^2}. \tag{12}$$

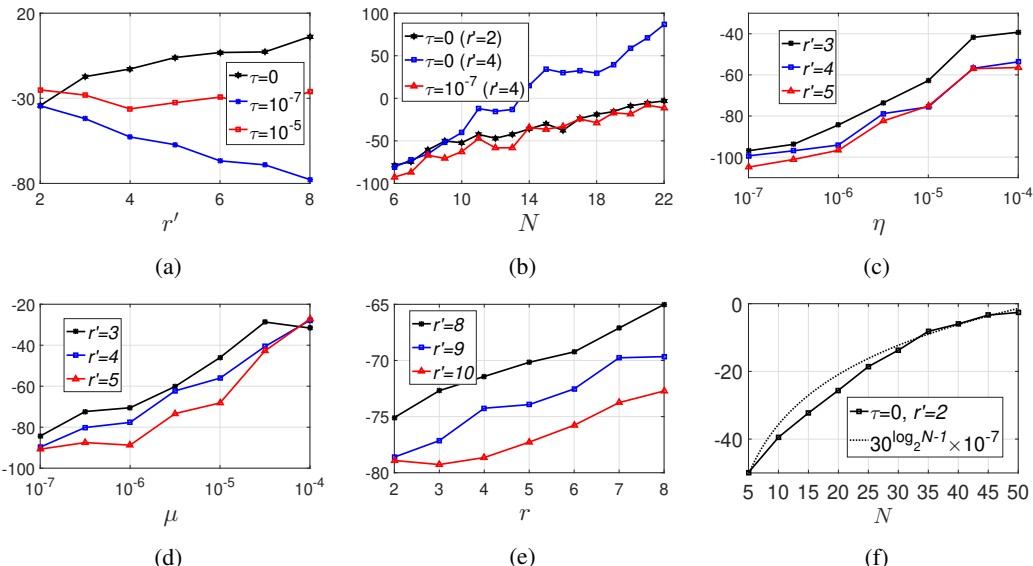

Figure 3: Performance (in MSE (dB) defined in (12)) of tensor cross approximations with truncated pseudo-inverse $\tau > 0$ and exact pseudo-inverse $\tau = 0$ for (a) different estimated rank $r'$ with $N = 10, r = 2, \mu = \eta = 10^{-5}$, (b) different tensor order $N$ with $r = 2, \mu = \eta = 10^{-5}$, (c) different level $\eta$ of deviation from TT format with $N = 10, r = 2, \mu = 10^{-5}$, (d) different noise level $\mu$ with $N = 10, r = 2, \eta = 10^{-5}$, (e) different rank $r$ with $N = 10, \mu = \eta = 10^{-5}$, (f) different tensor order $N$ with $r = 2, \mu = 0, \eta = 10^{-5}$.

In the first experiment, we set $N = 10$, $d = 2$, $r = 2$ and $\mu = \eta = 10^{-5}$ and compare the performance of tensor cross approximations with different $\tau$ and $r'$. From Fig. 3a, we observe that when $r'$ is specified as the exact rank (i.e., $r' = r$), all of the TT cross approximations are very accurate except when $\tau$ is relatively large and thus useful information is truncated. On the other hand, when $r' > r$, the performance of cross approximations with the exact pseudo-inverse ($\tau = 0$) degrades. This is because the intersection matrices $\boldsymbol{T}(I^{\leq k}, I^{>k})$ in (4) or $\widetilde{\boldsymbol{T}}(I^{\leq k}, I^{>k})$ in (9) become ill-conditioned. Fortunately, since the last $r' - r$ singular values of $\boldsymbol{T}(I^{\leq k}, I^{>k})$ and $\widetilde{\boldsymbol{T}}(I^{\leq k}, I^{>k})$ only capture information about the approximation error and measurement error, this issue can be addressed by using the truncated pseudo-inverse with suitable $\tau > 0$ which can make TT cross approximation stable. We notice that larger $r'$ gives better cross approximation due to more measurements.

In the second experiment, we test the performance of tensor cross approximations with different tensor orders $N$. For $N$ ranging from 6 to 22, we set all mode sizes $d = 2$ and $\mu = \eta = 10^{-5}$. Here $d = 2$ corresponds to a quantum state, and while $d = 2$ is small, the tensor is still huge for large $N$ (e.g., $N = 22$). We also set all ranks $r = 2$. From Fig. 3b, we can observe that the TT cross approximation with the pseudo-inverse is stable as $N$ increases when $r' = r$ (black curve) but becomes unstable when $r' > r$ (blue curve). Again, for the latter case when $r' > r$, the TT cross approximation achieves stable performance by using the truncated pseudo-inverse.

In the third experiment, we demonstrate the performance for different $\eta$, which controls the level of low-rank approximation error. Here, we set $\mu = 10^{-7}$, $r = 2$, $\tau = 10^{-2}\eta$, and consider the over-specified case where $r'$ ranges from 3 to 5. Fig. 3c shows that the performance of TT cross approximation with the truncated pseudo-inverse is stable with the increase of $\eta$ and different $r'$.

In the fourth experiment, we demonstrate the performance for different measurement error levels $\mu$. Similar to the third experiment, we set $\eta = 10^{-7}$, $r = 2$, $\tau = 10^{-2}\mu$, and consider the over-specified case where $r'$ ranges from 3 to 5. We observe from Fig. 3d that tensor train cross approximation still provides stable performance with increasing $\mu$ and different $r'$.

In the fifth experiment, we test the performance for different $r$. We set $\mu = \eta = 10^{-5}$, $d = 2$, $\tau = 10^{-2}\eta$ and consider the case where $r'$ ranges from 8 to 10. Fig. 3e shows that the peformance of TT cross approximation is stable with the increase of $r$, with recovery error in the curves roughly increasing as $O(r^3)$ which is consistent with Theorem 4. Note that in practice $r$ is often unknown a priori. For this case, the results in the above figures show that we can safely use a relatively large estimate $r'$ for tensor train cross approximation with the truncated pseudo-inverse.

In the final experiment, we test the performance with further larger tensor orders $N$. To achieve this goal, we work on tensors in the exact TT format (i.e., $\eta = 0$) since in this case we only need to store

the tensor factors with $O(Nr^2)$ elements rather than the entire tensor. We set $r' = r = 2$, $\mu = 10^{-5}$ and consider one case where $N$ varies from 5 to 50. We observe from Fig. 3f that the approximation error curve increases only logarithmically in terms of the order $N$, which is consistent with (10).

## 5   Conclusion and Outlook

While TT cross approximation has achieved excellent performance in practice, existing theoretical results only provide element-wise approximation accuracy guarantees. In this paper, we provide the first error analysis in terms of the entire tensor for TT cross approximation. For a tensor in an approximate TT format, we prove that the cross approximation is stable in the sense that the approximation error for the entire tensor does not grow exponentially with the tensor order, and we extend this guarantee to the scenario of noisy measurements. Experiments verify our results.

An important future direction we will pursue is to study the implications of our results for quantum tomography. In particular, we aim to find out whether our results imply that the statistical errors encountered in quantum measurements will not proliferate if we perform cross approximation of the quantum many-body state. Another future direction of our work is to design optimal strategies for the selections of the tensor elements to be measured. One possible approach is to extend methods developed for matrix cross approximation [60], and another approach is to incorporate leverage score sampling [76–78]. It is also interesting to use local refinement to improve the performance of TT cross approximation. We leave these explorations to future work.

## Acknowledgment

We acknowledge funding support from NSF Grants No. CCF-1839232, PHY-2112893, CCF-2106834 and CCF-2241298, as well as the W. M. Keck Foundation. We also thank the four anonymous reviewers for their constructive comments.

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
