# OpenReview forum: "Error Analysis of Tensor-Train Cross Approximation"
_NeurIPS.cc/2022/Conference — NeurIPS 2022 Accept_

### Official Review · Reviewer_oKWt · 2022-07-10

**Rating:** 7
**Confidence:** 3
**Soundness:** 4 excellent
**Presentation:** 4 excellent
**Contribution:** 3 good

**Summary:**

This paper provides the error analysis for the entire tensor, given clean or noisy measurements. Furthermore, it sheds light on how to sample the tensors for better approximation error. The results also show that the approximation error may not grow exponentially with the order of the tensor, implying more possible usefulness in quantum applications.

**Questions:**

I believe that the authors have already sufficiently answered my questions in the paper, including the error analysis in the presence of noises and the gap between an optimal cross approximation and any approximation. It would be better if the author can provide an example showing how Theorem 3 helps in selecting rows and columns to sample. While the authors have listed the optimal strategies for selections as a future direction, it would be nice if the authors can give some intuitions at a high level. It would help the readers better assess the theoretical implication and usefulness. Also, could the authors provide some intuitions of why the truncated pseudo-inverse could lead to better performance than the (pseudo-)inverse for TT cross approximation? Is it just because of the conditioning numerical stabilities?


**Limitations:**

NA.

**Strengths And Weaknesses:**

This paper is well-supported by theoretical results as well as numerical experiments.

It provides the first error analysis for TT cross approximation that goes beyond element-wise bounds. The significance herein is that plainly applying element-wise results will yield an exponential growth in errors, and may imply a false sense of lack of accuracy.
The authors have done a good job walking readers through the motivation, definitions, and theoretical results. The technical contribution is self-evident.

---

> ### Author Response · Authors · 2022-08-02
> **Response to Reviewer oKWt**
>
> We thank the reviewer for the appreciation of our work, as well as the detailed and thoughtful comments. We address them in detail as follows.
>
> > Comment: It would be better if the author can provide an example showing how Theorem 3 helps in selecting rows and columns to sample… help the readers better assess the theoretical implication and usefulness.
>
> Our reply: Thanks for the suggestion. Theorem 3 implies that one can select rows and columns that give small quantities $a,b,c$. A practical algorithm to achieve this is the greedy restricted cross interpolation (GRCI) algorithm [A].  We may also exploit derandomized algorithms [B] to find interpolation sets that could potentially yield small quantities $a,b,c$. We have included this discussion right after Theorem 3 in the revised manuscript.
>
> [A] D. V. Savostyanov. Quasioptimality of maximum-volume cross interpolation of tensors. Linear Algebr. Appl., 458(1):217–244, Oct. 2014.
>
> [B] A. Cortinovis and D. Kressner. Low-rank approximation in the Frobenius norm by column and row subset selection. SIAM J. Matrix Anal. Appl., 41(4):1651–1673, Nov. 2020.
>
> > Comment: Intuitions of why the truncated pseudo-inverse could lead to better performance than the (pseudo-)inverse for TT cross approximation? Is it just because of the conditioning numerical stabilities?
>
> Our reply: The reviewer’s intuition is correct. We observe from Figure 3 that when the rank is exactly specified, then the pseudo-inverse and truncated pseudo-inverse give similar performance. But when the rank is overspecified, i.e., $r’>r$, the intersection matrix becomes ill-conditioned, or the last $r’ - r$ singular values mainly contain the information from the approximation error and measurement error. This issue can be addressed by the truncated pseudo-inverse which leads to better performance. We incorporate this discussion in Section 4 of the revised manuscript.

---

> > ### Comment · Reviewer_oKWt · 2022-08-07
> > **Thank the authors for the reply**
> >
> > I would like to thank the authors for their response. I have read the threads and I tend to keep my score.
> >
> > Particularly, I checked the authors' response to reviewer iCWy and believed the concerns should have been addressed. If not, it would be helpful if reviewer iCWy can give some pointers on what other existing bounds should be furthered added for comparisons.

---

> > > ### Author Response · Authors · 2022-08-08
> > > **Thank you**
> > >
> > > We thank the reviewer for taking time to look at the response and revision and for the appreciation of our efforts.

---

### Official Review · Reviewer_HshM · 2022-07-11

**Rating:** 5
**Confidence:** 1
**Soundness:** 2 fair
**Presentation:** 2 fair
**Contribution:** 2 fair

**Summary:**

Tensor train (TT) application is a feasible approach to generalize tensor information. This paper discussed the cross approximation error for TT decomposition, where it generalize the upper bound for the decomposition error of the entire tensor and proved the cross approximation  is stable that does not grow exponentially with tensor order.

**Questions:**

Please see above.

**Ethics Review Area:**

["I don’t know"]

**Strengths And Weaknesses:**

Strengths:
1. This paper orchestrated a comprehensive proof toward the tensor train decomposition with cross approximation. Both tensor train decomposition and cross approximation are well-known techniques in the field.
2. Experimental analysis revealed that the error of the decomposition is stable with the increase of tensor order.
3. The author also introduce the tensor train decomposition and cross approximation in detail and illustrated their differences with related methods.

Weaknesses:
1. Full disclosure, I am not an expert in this field. On top of everything, I am not able to evaluate the significance of the proof. Somehow, I feel it is rather intuitive that error did not grow exponentially is because the decomposition scheme of the tensor train decomposition. I.e., irrespective of the actual decomposition method, i.e.cross approximation in this paper, the tensor train decomposition scheme will restrain the exponential grow of the error.
2. I also have a hard time understand the proof. Staring with the proof of theorem 1. Does it mean, as long as rank of U and A are the same, the column space of R can be represented by U. And more confusing to me, C U R are all retrieved from A itself. I understand if two matrices have the same rank, we probably can have same set of columns or rows to represent the column and row space of these two matrices. But does it still hold, when all the decomposition matrices, C U R are all retrieved from the same matrix A? I am not very sure about this.
3. Does the tensor train decomposition robust towards permutation? It looks like what the decomposition doing is to fold the tensor into a big matrix. During the process, it will lose track of the structure of the tensor. While permutating row, column, order of the tensor, does the decomposition result will be the same?

---

> ### Author Response · Authors · 2022-08-02
> **Response to Reviewer HshM**
>
> We thank the reviewer for the appreciation of our work, as well as the detailed and thoughtful comments. We address them in detail as follows.
>
> > Comment: I feel it is rather intuitive that error did not grow exponentially is because the decomposition scheme of the tensor train decomposition
>
> Our reply: As explained in [A,B], due to the cascaded multiplications between the factors, the approximation error in tensor-train cross approximation could conceivably grow exponentially with the order; this is actually proved to be the case for Tucker tensors. As in [A,B], we bound the approximation error by exploiting the tensor train structures, but for the reason above, we do not find the result to be obvious. In the revision, we have now included this discussion in the introduction. In case the reviewer has further intuition which we have not captured, we would be happy to include it in the revision.
>
> [A] D. V. Savostyanov. Quasioptimality of maximum-volume cross interpolation of tensors. Linear Algebr. Appl., 458(1):217–244, Oct. 2014.
>
> [B] A. Osinsky. Tensor trains approximation estimates in the chebyshev norm. Comp. Math. Math. Phys., 59(2):201–206, May 2019.
>
> > Comment: I also have a hard time understand the proof. Staring with the proof of theorem 1. Does it mean, as long as rank of U and A are the same, the column space of R can be represented by U. And more confusing to me, C U R are all retrieved from A itself. I understand if two matrices have the same rank, we probably can have same set of columns or rows to represent the column and row space of these two matrices. But does it still hold, when all the decomposition matrices, C U R are all retrieved from the same matrix A? I am not very sure about this.
>
> Our reply: To explain the reviewer’s concern,since $R$ contains some rows from $A$, we may write $A$ as $A = [R; B]$. Likewise, we can further write $R$ as $R = [U \ V]$. Now it is clear that $rank(U)\le rank(R) \le rank(A)$. Thus, if $A$ and $U$ have the same rank, then $R$ and $U$ have the same rank. Therefore, $R$ and $U$ have the same column space since otherwise $R$ must have larger rank than $U$. Theorem 1 mainly serves as an introduction to cross approximation. More detailed description can be found in [C]. We have revised the proof of Theorem 1 based on the above discussion. We have also revised the other proofs in the Appendix.
>
> [C] K. Hamm and L. Huang. Perspectives on cur decompositions. Appl. Comput. Harmon. Anal., 48(3):1088– 1099, May 2020.
>
> > Comment: While permutating row, column, order of the tensor, will the tensor train decomposition result be the same?
>
> Our reply: Thanks for this interesting question. According to the TT decomposition in Eq. (3), permuting the rows within each index $i_j$ will give a similar decomposition. But in general the decomposition is not invariant to the permutation of the order of tensors due to its strictly sequential multilinear products over the latent cores. This is partially addressed by the tensor ring decomposition [D] which is invariant to circular dimensional permutation. We have included this discussion right after Eq. (3).
>
> [D] Qibin Zhao, Guoxu Zhou, Shengli Xie, Liqing Zhang, and Andrzej Cichocki. Tensor ring decomposition. arXiv preprint arXiv:1606.05535, 2016.

---

### Official Review · Reviewer_iCWy · 2022-07-12

**Rating:** 3
**Confidence:** 3
**Soundness:** 3 good
**Presentation:** 3 good
**Contribution:** 2 fair

**Summary:**

This paper considers the problem of tensor train decomposition. It provides theoretical guarantees of cross approximation in terms of the Frobenius norm for both exact and noisy measurements. Numerical experiments are included to validate the theories.

**Questions:**

1. In theorem 2, it seems that the upper bound (6) will blow up if $\kappa$ is a constant, which happens when $\kappa$ increases with $d_1, \dots d_N$ and $N$. In this case, the current bound may be worse than that obtained by directly multiplying the entrywise bound by $\sqrt{d_1 \dots d_n}$. Authors should include more discussions about $\kappa$. In addition, as discussed in "Weakness", I think authors should compare the bounds with existing works, and this also applies to Theorems 3-4.

2. As one can expect, the notations will be messy in tensor papers. It would make it more accessible to readers if authors can summarize the key notations in a section.

**Strengths And Weaknesses:**

Strength:
Overall, the paper is easy to read and the literature review is comprehensive. Under certain conditions, the theoretical guarantees improve upon the prior art.

Weakness:
As stated in the abstract and introduction, the main motivation of this work is that the existing element-wise approximation accuracy guarantees lead to loose bounds. However, the current paper only shows its own upper bounds and there is no comparison with existing results or minimax lower bounds. As a result, it is not obvious to me if it indeed leads to better theoretical guarantees. I believe it would largely improve the quality of the paper either by comparing the results with the prior works both theoretically and numerically, or by establishing the matching lower bounds to show the tightness of the upper bounds.

---

> ### Author Response · Authors · 2022-08-02
> **Response to the Reviewer iCWy**
>
> We thank the reviewer for the thoughtful comments. We address them in detail as follows.
>
> > Comment: “As stated in the abstract and introduction, the main motivation of this work is that the existing element-wise approximation accuracy guarantees lead to loose bounds. However, the current paper only shows its own upper bounds and there is no comparison with existing results or minimax lower bounds. As a result, it is not obvious to me if it indeed leads to better theoretical guarantees. I believe it would largely improve the quality of the paper either by comparing the results with the prior works both theoretically and numerically, or by establishing the matching lower bounds to show the tightness of the upper bounds.”
> *and*  "Authors should include more discussions about $\kappa$. In addition, as discussed in "Weakness", I think authors should compare the bounds with existing works, and this also applies to Theorems 3-4.”
>
> Our response:  Thanks for these comments. We have included a discussion of the role of $\epsilon$ as a lower bound in the paragraphs above and below Theorem 2 in the revision.
>
> We did not compare our bounds with the existing ones [A,B] since they only concern the element-wise error (Chebyshev norm), while our results are about the entire tensor (Frobenius norm). But following the reviewer’s suggestion, we now present the result from [A, Theorem 1] which actually shares a similar form as our upper bound Eq. (7) (which was Eq. (6) in the previous version) and involves a similar condition number $\kappa$. It is then obvious to the readers that the previous error bound on element-wise error can be exponentially looser than our bound. Moreover, we also emphasize that our results contain an additional improvement over existing results in that the parameter $\kappa$ in [A] can be out of control while we upper bound it in our Theorem 3.
>
> Following the reviewer's suggestion,  we have performed two additional numerical experiments in the revised paper that further compare our error bounds with the actual tensor-train cross approximation error. The results show that our bounds are indeed tight at least qualitatively.
>
> [A] D. V. Savostyanov. Quasioptimality of maximum-volume cross interpolation of tensors. Linear Algebr. Appl., 458(1):217–244, Oct. 2014.
>
> [B] A. Osinsky. Tensor trains approximation estimates in the chebyshev norm. Comp. Math. Math. Phys., 59(2):201–206, May 2019.
>
> > Comment: “As one can expect, the notations will be messy in tensor papers. It would make it more accessible to readers if authors can summarize the key notations in a section.”
>
> Our reply: We agree with the reviewer that in general the notations for tensors could be messy and more complicated than for matrices. This is the reason that we have summarized the key notations for tensors at the beginning of Section 3 and introduced the notations for tensor train decomposition in section 3.1 and for TT cross approximation in section 3.2 when they appear. Summarizing all these definitions will require substantial space. So we will summarize them if we are allowed an additional content page for the camera-ready version as in previous years.

---

### Official Review · Reviewer_ViNA · 2022-07-12

**Rating:** 6
**Confidence:** 4
**Soundness:** 3 good
**Presentation:** 3 good
**Contribution:** 2 fair

**Summary:**

In this paper, the error of tensor train decomposition via cross approximation is analyzed. The cross approximation of tensor train decomposition is an analog of CUR decomposition for a higher-order tensor. The authors derive an error bound in terms of the Frobenius norm where the observation noise is taken into account. Numerical experiments are conducted to evaluate the actual dependency between the error and other parameters such as noise level and rank.

**Questions:**

### Question 1
According to Lines 61--62 in Introduction, the main motivation to evaluate the error based on the Frobenius norm is that the prior studies only provide "element-wise error bound" and "when one extends these element-wise results to the entire tensor, the Frobenius norm error bound grows exponentially with the order of the tensor". However, the derived bound (6) contains \epsilon, which is the Frobenius norm between observed and approximated tensors. For example, if each element of the residual tensor follows standard normal, \epsilon becomes a square root of a xi-square variable, and its mean value increases exponentially w.r.t. the order of the tensor. In the end, the derived bounds do not resolve exponential dependence. If I misunderstood something, please correct me.

### Question 2
In Introductions, several applications of tensor train decomposition and CUR matrix decomposition. However, as far as noticed, no specific application of "cross approximation (CUR) of tensor train decomposition" is provided. Could you provide a few examples for that, especially the applications in the machine learning literature?

### Question 3
I'm not convinced that the experiments of Figure 1 prove the correctness of the derived bound (6). For example, (b) doesn't exhibit the logarithmic curve w.r.t. N and (a) doesn't show the linearity in terms of r. I suspect this is because the problem size is not large enough. As a sanity check, I highly recommend conducting experiments on a larger scale so that we can clearly see the derived relationships numerically.

### Question 4
The norm that appears in the definition of \kappa seems undefined.

**Limitations:**

See Questions above.

**Strengths And Weaknesses:**

Strengths
1. A technically solid result is provided.
1. Mathematical notation is consistently used and easy to follow.

Weaknesses
1. The motivation to develop the Frobenius norm-based analysis is ambiguous. See Question 1.
2. The scope of this study (cross approximation of tensor train decomposition) seems a bit limited to the NeurIPS community. See Question 2.
3. The derived bound is not fully examined in the numerical experiments. See Question 3.

---

> ### Author Response · Authors · 2022-08-02
> **Response to Reviewer ViNA**
>
> We thank the reviewer for the appreciation of our work, as well as the detailed and thoughtful comments. We address them in detail as follows.
>
> > Comment:  However, the derived bound (6) contains \epsilon, which is the Frobenius norm between observed and approximated tensors. For example, if each element of the residual tensor follows standard normal, \epsilon becomes a square root of a xi-square variable, and its mean value increases exponentially w.r.t. the order of the tensor. In the end, the derived bounds do not resolve exponential dependence. If I misunderstood something, please correct me.”
>
> Our reply: We first note that $\epsilon$ defined in Eq. (5) represents the approximation error of the original tensor in the TT format. As we now clarify in Eq. (6) of the revised paper, this $\epsilon$ represents a fundamental lower bound on the approximation performance of any low-rank TT approximation, including one formed by cross approximation. Fortunately, tensors in practical applications such as image and video and quantum states can be well represented in TT format with very small $\epsilon$. A dependence on $\epsilon$ was included in previous element-wise cross approximation error bounds. Our goal is to derive Frobenius-norm for TT cross approximation in which $\epsilon$ is not amplified exponentially. We have included this discussion above and below Theorem 2 in the revision.
>
> On the other hand, when the measurements are noisy, the bound in Theorem 4 involves the energy of the noise. In this case, as the reviewer said, if the noise follows a standard normal distribution, then the energy becomes a square root of a chi-square variable, and its mean value increases w.r.t. the order of tensor $N$. However, this scaling is actually only linear, not exponential, since we only need to sample $O(Nr^2)$ measurements. We highlight this after Theorem 4.
>
> > Comment: “In Introductions, several applications of tensor train decomposition and CUR matrix decomposition. However, as far as noticed, no specific application of "cross approximation (CUR) of tensor train decomposition" is provided. Could you provide a few examples for that, especially the applications in the machine learning literature?”
>
> Our reply: Thank you for the suggestion. TT cross approximation has been used for machine learning applications like learning low-rank embeddings of visual data [A], density estimation [B], surrogate visualization modeling [C], etc. It can also be used for many other applications like image and video compression [D] where it is desirable to only sample a few elements rather than the entire tensor. We have included the above discussion in the introduction and Section 3.2 of the revision.
>
> [A] M. Usvyatsov, A. Makarova, R. Ballester-Ripoll, M. Rakhuba, A. Krause, and K. Schindler. Cherry-picking gradients: Learning low-rank embeddings of visual data via differentiable cross-approximation. In Proc. IEEE Int. Conf. Comput. Vis., pages 11426–11435, 2021.
>
> [B] Andrei Chertkov and Ivan Oseledets. Solution of the fokker–planck equation by cross approximation method in the tensor train format. Frontiers in Artificial Intelligence, 4, 2021.
>
> [C] R. Ballester-Ripoll, E. G. Paredes, and R. Pajarola. A surrogate visualization model using the tensor train format. In Proc. SIGGRAPH ASIA 2016 Symposium on Visualization, pages 1–8, 2016.
>
> [D] Johann A Bengua, Ho N Phien, Hoang Duong Tuan, and Minh N Do. Efficient tensor completion for color image and video recovery: Low-rank tensor train. IEEE Transactions on Image Processing, 26(5):2466–2479, 2017.
>
> > Comment: I'm not convinced that the experiments of Figure 1 prove the correctness of the derived bound (6) ... I highly recommend conducting experiments on a larger scale so that we can clearly see the derived relationships numerically.
>
> Our reply: We first note that Figure 3(a) plots the performance for different estimated ranks $r’$ with a fixed rank-$r$ tensor. It shows the stable performance of TT cross approximation with truncated pseudo-inverse even when the rank $r’$ is overspecified. Following the reviewer’s suggestion, we have included additional experiments in terms of $r$ in Figure 3(e), and the recovery error curve increases as $O(r^3)$, which is consistent with Theorem 3. For Figure 3(b), the tensor has $2^{N}$ elements so it already has a relatively large size when, say, $N = 22$. To further increase the size, we have done additional numerical experiments on tensors in the exact TT format which allows us to increase $N$ without storing every tensor element explicitly. The result is plotted in Figure 4. We indeed observe a logarithmic curve with respect to $N$, which is consistent with Theorem 4.
>
> > Comment: “The norm that appears in the definition of \kappa seems undefined.”
>
> Our reply: Thank you for pointing out this oversight by us. It refers to the spectral norm. We have added the definition at the end of the introduction.

---

### Meta-Review · Area_Chair_sksJ · 2022-08-27

**Recommendation:** Accept
**Confidence:** Certain

**Metareview:**

The reviewers agree that the theoretical result shown in the paper improves upon prior art and that the experiments are compelling.

All comments and concerns of the reviewers have been well addressed by the authors.


**Award:**

No

---

### Decision · Program_Chairs · 2022-09-14

Accept